# Learning Neuron Non-Linearities with Kernel-Based Deep Neural Networks

## Abstract

The effectiveness of deep neural architectures has been widely supported in terms of both experimental and foundational principles. There is also clear evidence that the activation function (e.g. the rectifier and the LSTM units) plays a crucial role in the complexity of learning. Based on this remark, this paper discusses an optimal selection of the neuron non-linearity in a functional framework that is inspired from classic regularization arguments. A representation theorem is given which indicates that the best activation function is a kernel expansion in the training set, that can be effectively approximated over an opportune set of points modeling 1-D clusters. The idea can be naturally extended to recurrent networks, where the expressiveness of kernel-based activation functions turns out to be a crucial ingredient to capture long-term dependencies. We give experimental evidence of this property by a set of challenging experiments, where we compare the results with neural architectures based on state of the art LSTM cells.

## 1 Introduction

By and large, the appropriate selection of the activation function in deep architectures is regarded as an important choice for achieving challenging performance. For example, the rectifier function (Glorot et al., 2011) has been playing an important role in the impressive scaling up of nowadays deep nets. Likewise, LSTM cells (Hochreiter & Schmidhuber, 1997) are widely recognized as the most important ingredient to face long-term dependencies when learning by recurrent neural networks. Both choices come from insightful ideas on the actual non-linear process taking place in deep nets. At a first glance, one might wonder why such an optimal choice must be restricted to a single unit instead of extending it to the overall function to be learned. In addition, this general problem has been already been solved; its solution (Poggio & Girosi, 1990; Girosi et al., 1995; 2000) is in fact at the basis of kernel machines whose limitations as shallow nets have been widely addressed (see e.g. (LeCun et al., 2015; Mhaskar et al., 2016)). However, the optimal formulation given for the neuron non-linearity enjoys the tremendous advantage of acting on 1-D spaces. This strongly motivates the reformulation of the problem of learning in deep neural network as a one where the weights and the activation functions are jointly determined by optimization in the framework of regularization operators (Smola et al., 1998), that are used to enforce the smoothness of the solution.

The idea of learning the activation function is not entirely new. In (Turner & Miller, 2014), activation functions are chosen from a pre-defined set and combine this strategy with a single scaling parameter that is learned during training. It has been argued that one can think of this function as a neural network itself, so the overall architecture is still characterized by a directed acyclic graph (Castelli & Trentin, 2014). Other approaches learn activation functions as piecewise linear (Agostinelli et al., 2014), doubled truncated gaussian (Su et al., 2017) or Fourier series (Eisenach et al., 2016).

While working on this representational issues we have recently discovered the paper by (Scardapane et al., 2017), which introduces a family of activation functions that are based on a kernel expansion at every neuron. The proposed approach is based on the nice intuition that a kernel-based representation at for the neuron function is computationally efficient, yet very effective in terms of representation. The authors provide strong support to their idea by significant experimental results. Interestingly, in this paper, where we provide additional independent support to the kernel-based representation of the neuron function given in (Scardapane et al., 2017) by reinforcing the idea in different ways. In particular, we prove that, like for kernel machines, the optimal solution of the variational problem that

characterizes the process of supervised learning in the framework of regularization can be expressed by a kernel expansion, so as the overall optimization is reduced to the discovery of a finite set of parameters. The risk function to be minimized contains the weights of the network connections, as well as the parameters associated with the the points of the kernel expansion. Hence, the classic learning of the weights of the network takes place with the concurrent development of the optimal shape of the activation functions, one for each neuron. As a consequence, the machine architecture turns out to enjoy the strong representational issues of deep networks in high dimensional spaces that is conjugated with the elegant and effective setting of kernel machines for the learning of the activation functions. The powerful unified regularization framework is not the only feature that emerges from the proposed architecture. Interestingly, unlike most of the activation functions used in deep networks, those that are typically developed during learning, are not necessarily monotonic. This property has a crucial impact in their adoption in classic recurrent networks, since this properly addresses classic issues of gradient vanishing when capturing long-term dependencies. Throughout this paper, recurrent networks with activation functions based on kernel expansion, are referred to as Kernel-Based Recurrent Networks (KBRN). The intuition is that the associated iterated map can either be contractive or expansive. Hence, while in some states the contraction yields gradient vanishing, in others the expansion results in to gradient pumping, which allows the neural network to propagate information back also in case of long time dependences. The possibility of implementing contractive and expanding maps during the processing of a given sequence comes from the capabilities of KBRN to develop different activation functions for different neurons that are not necessarily monotonic. This variety of units is somewhat related to the clever solution proposed in LSTM cells (Hochreiter & Schmidhuber, 1997), where the authors realized early that there was room for getting rid of the inherent limitation of the contractive maps deriving from sigmoidal units. This contribution to the representation and learning in recurrent neural network is another difference with respect to the related contribution given in (Scardapane et al., 2017).

The given experimental results provided below demonstrate this property on challenging benchmarks that are inspired from seminal paper (Bengio et al., 1993), where the distinctive information for classification of long sequences is only located in the first positions, while the rest contains uniformly distributed noisy information. We get very promising results on these benchmarks when comparing KBRN with state of the art recurrent architectures based on LSTM cells.

## 2 REPRESENTATION AND LEARNING

The feedforward architecture that we consider is based on a directed graph $D \sim (V, A)$, where $V$ is the set of ordered vertices and $A$ is the set of the oriented arcs. Given $i, j \in V$ there is connection from $i$ to $j$ iff $i \prec j$. Instead of assuming a uniform activation function for each vertex of $D$, a specific function $f$ is attached to each vertex. We denote with $I$ the set of input neurons, with $O$ the set of the output neurons and with $H = V \setminus (I \cup O)$ the set of hidden neurons; the cardinality of these sets will be denoted as $|I|, |O|, |H|$ and $|V| \equiv n$. Without loss of generality we will also assume that: $I = \{1, 2, \ldots, |I|\}$, $H = \{|I| + 1, |I| + 2, \ldots, |I| + |H|\}$ and $O = \{|I| + |H| + 1, |I| + |H| + 2, \ldots |I| + |H| + |O|\}$.

The learning process is based on the training set $T_N = \{ (e^\kappa, y^\kappa) \in \mathbb{R}^{|I|} \times \mathbb{R}^{|O|} \mid \kappa = 1, \ldots N \}$. Given an input vector $z = (z_1, z_2, \ldots z_{|I|})$, the output associated with the vertices of the graph is computed as follows[1]:

$$x_i(z) = z_i (i \in I) + f_i(a_i)(i \notin I), \qquad (1)$$

with $a_i = \sum_{j \in \mathrm{pa}(i)} w_{ij} x_j + b_i$, where $\mathrm{pa}(i)$ are the parents of neuron $i$, and $f_i \colon \Omega_\Lambda \to \mathbb{R}$ are one dimensional real functions; $\Omega_\Lambda := [-\Lambda, \Lambda]$, with $\Lambda$ chosen big enough, so that Eq. (1) is always well defined. Now let $f = (f_1, f_2, \ldots, f_n)$ and define the output function of the network $F(\cdot, w, b; f) \colon \mathbb{R}^{|I|} \to \mathbb{R}^{|O|}$ by

$$F_i(z, w, b; f) := x_{i+|I|+|H|}(z), \quad i = 1, \ldots, |O|.$$

The learning problem can then be formulated as a double optimization problem defined on both the weights $w, b$ and on the activation functions $f_i$. It is worth mentioning that while the optimization on the weights of the graph reflects all important issues connected with the powerful representational

---

[1]We use Iverson's notation: Given a statement $A$, we set $(A)$ to 1 if $A$ is true and to 0 if $A$ is false

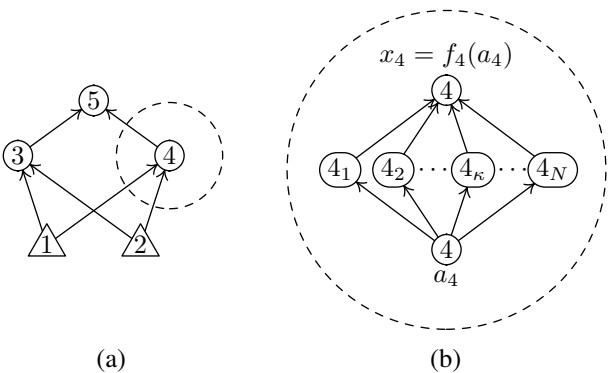

(a)          (b)

Figure 1: (a) A simple network architecture; the output evaluated using Eq. (1) is $x_5(z_1, z_2) = f_5(w_{53}f_3(w_{31}z_1 + w_{32}z_2 + b_3) + w_{54}f_4(w_{41}z_1 + w_{42}z_2 + b_4) + b_5)$. (b) Highlight of the structure of neuron $4$ (encircled in the dashed line) of (a): The activation function $f_4$ of the neuron is computed as an expansion over the training set. Each neuron $4_j$, $j = 1, \ldots, N$ in the figure corresponds to the term $g(a_4 - a_4^j)$ in Eq. (4).

properties of deep nets, the optimal discovery of the activation functions are somewhat related to the framework of kernel machines. Such an optimization is defined with respect to the following objective function:

$$E(f; w, b) := \frac{1}{2}\sum_{i=1}^{n}(Pf_i, Pf_i) + \sum_{\kappa=1}^{N}V(e^\kappa, y^\kappa, F(e^\kappa, w, b; f)), \tag{2}$$

which accumulates the empirical risk and a regularization term (Smola et al., 1998). Here, we indicate with $(\cdot, \cdot)$ the standard inner product of $L^2(\Omega_\Lambda)$, with $P$ a differential operator of degree $p$, while $V$ is a suitable loss function.

Clearly, one can optimize $E$ by independently checking the stationarity with respect to the weights associated with the neural connections and the stationarity with respect to the activation functions. Now we show that the stationarity condition of $E$ with respect to the functional variables $f$ (chosen in a functional space $X_p$ that depends on the order of differential operator $P$) yields a solution that is very related to classic case of kernel machines that is addressed in (Smola et al., 1998). If we consider a variation $v_i \in C_c^\infty(\Omega_\Lambda)$ with vanishing derivatives on the boundary [2] of $\Omega_\Lambda$ up to order $p - 1$ and define $\varphi_i(t) := E(f_1, \ldots, f_i + tv_i, \ldots, f_n; w, b)$. The first variation of the functional $E$ along $v_i$ is therefore $\varphi_i'(0)$. When using arguments already discussed in related papers (Poggio & Girosi, 1990; Girosi et al., 1995; Smola et al., 1998) we can easily see that

$$\varphi_i'(0) = \int_{\Omega_\Lambda}\left(Lf_i(a) + \sum_{\kappa=1}^{N}\alpha_i^\kappa\delta_{a_i^\kappa}(a)\right)v_i(a)\, da,$$

where $\alpha_i^\kappa = \nabla_F V \cdot \partial_{f_i}F$ and $L = P^*P$, $P^*$ being the adjoint operator of $P$. We notice in passing that the functional dependence of $E$ on $f$ is quite involved, since it depends on the compositions of linear combinations of the functions $f_i$ (see Figure 1–(a)). Hence, the given expression of the coefficients $\alpha_i^\kappa$ is rather a formal equation that, however, dictates the structure of the solution.

The stationarity conditions $\varphi_i'(0) = 0$ reduce to the following Euler-Lagrange (E-L) equations

$$Lf_i(a) + \sum_{\kappa=1}^{N}\alpha_i^\kappa\delta_{a_i^\kappa}(a) = 0, \quad i = 1 \ldots n, \tag{3}$$

where $a_i^\kappa$ is the value of the activation function on the $\kappa$-th example of the training set. Let $g$ be the Green function of the operator $L$, and let $k$ BE the solution of $Lk = 0$. Then, we can promptly see that

$$f_i(a) = k(a) - \sum_{\kappa=1}^{N}\alpha_i^\kappa g(a - a_i^\kappa) \tag{4}$$

---

[2]We are assuming here that the values of the functions in $X_p$ at the boundaries together with the derivatives up to order $p - 1$ are fixed.

is the general form of the solution of Eq. (3). Whenever $L$ has null kernel, then this solution is reduced to an expansion of the Green function over the points of the training set. For example, this happens in the case of the pseudo differential operator that originates the Gaussian as the Green function. If we choose $P = d/dx$, then $L = -d^2/dx^2$. Interestingly, the Green function of the second derivative is the rectifier $g(x) = -\frac{1}{2}(|x| + x)$ and, moreover, we have $k(x) = mx + q$. In this case

$$f_i(a) = \theta_i a + \nu_i + \frac{1}{2} \sum_{\kappa=1}^{N} \alpha_i^\kappa |a - a_i^\kappa|, \tag{5}$$

where $\theta_i = m + \frac{1}{2} \sum_{\kappa=1}^{N} \alpha_i^\kappa$, while $\nu_i = q - \frac{1}{2} \sum_{\kappa=1}^{N} \alpha_i^\kappa a_i^\kappa$. Because of the representation structure expressed by Eq. (4), the objective function of the original optimization problem collapses to a standard finite-dimensional optimization on[3]

$$\hat{E}(\alpha, w, b) := E\Big(k(a) - \sum_\kappa \alpha^\kappa g(a - a^\kappa); w, b\Big) = R(\alpha) + \sum_{\kappa=1}^{N} V(e^\kappa, y^\kappa, \hat{F}(e^\kappa, w, b; \alpha));$$

here $R(\alpha)$ is the regularization term and $\hat{F}(e^\kappa, w, b; \alpha) := F\big(e^\kappa, w, b; k(a) - \sum_\kappa \alpha_i^\kappa g(a - a_i^\kappa)\big)$. This collapse of dimensionality is the same which leads to the dramatic simplification that gives rise to the theory of kernel machines. Basically, in all cases in which the Green function can be interpreted as a kernel, this analysis suggests the neural architecture depicted in Figure 1, where we can see the integration of graphical structures, typical of deep nets, with the representation in the dual space typical of kernel methods.

We can promptly see that the idea behind kernel-based deep networks can be extended to cyclic graphs, that is to recurrent neural networks. In that case, the analogous of Eq. (1) is:

$$h_i^{t+1} = f_i(a_i^{t+1}); \qquad a_i^{t+1} = b_i + \sum_{j \in \mathrm{pa}_{t \to t+1}(i)} w_{ij} h_j^t + \sum_{j \in \mathrm{pa}_{t+1}(i)} u_{ij} x_j^{t+1}.$$

Here we denote with $x_i^t$ the input at step $t$ and with $h_i^t$ the state of the network. The set $\mathrm{pa}_{t \to t+1}(i)$ contains the vertices $j$ that are parents of neuron $i$; the corresponding arcs $(j, i)$ are associated with a delay, while $\mathrm{pa}_t(i)$ vertices $j$ with non-delayed arcs $(j, i)$. The extension of learning in KBDNN to the case of recurrent nets is a straightforward consequence of classic Backpropagation Through Time.

## 3 APPROXIMATION AND ALGORITHMIC ISSUES

The actual experimentation of the model described in the previous section requires to deal with a number of important algorithmic issues. In particular, we need to address the typical problem associated with the kernel expansion over the entire training set, that is very expensive in computational terms. However, we can early realize that KBDNNs only require to express kernel in 1-D, which dramatically simplify the kernel approximation. Hence, instead of expanding $f_i$ over the entire training set, we can use a number of points $d$ with $d \ll N$. This means that the expansion in Eq. (4) is approximated as follows

$$f_i(a) \approx k(a) - \sum_{k=1}^{d} \chi_i^k g(a - c_i^k), \tag{6}$$

where $c_i^k$ and $\chi_i^k$ are the centers and parameters of the expansion, respectively. Notice that $\chi_i^k$ are replacing $\alpha_i^\kappa$ in the formulation given in Section 2). We consider $c_i^k$ and $\chi_i^k$ as parameters to be learned, and integrate them in the whole optimization scheme.

In the experiments described below we use the rectifier (ReLU) as Green function ($g(x) = -\frac{1}{2}(|x| + x)$) and neglect the linear terms from both $g(x)$ and $k(x)$. We can easily see that this is compatible with typical requirements in machine learning experiments, where in many cases the expected solution is not meaningful with very large inputs. For instance, the same assumption is typically at the basis of kernel machines, where the asymptotic behavior is not typically important. The regularization term

---

[3]Here we omit the dependencies of the optimization function from the parameters that defines $k$.

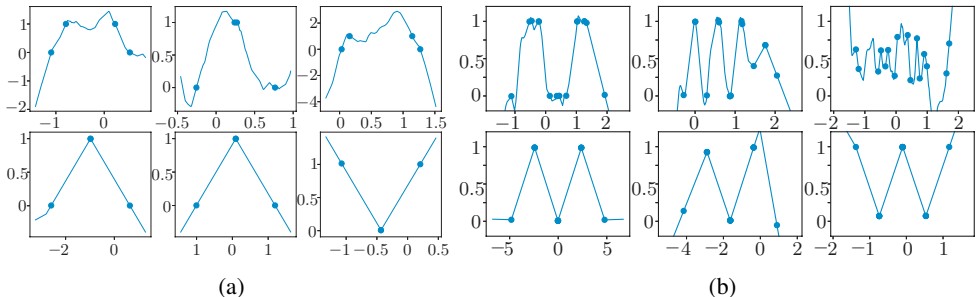

Figure 2: **XOR.** The plots show the activation functions learned by the simplest KBDN which consists of one unit only for the 2-dim (2a) and 4-dim (2b) XOR. The first/second row refer to experiments with without/with regularization, whereas the three columns correspond with the chosen number of point for the expansion of the Green function $d = 50, 100, 300$.

$R(\chi)$ can be inherited from the regularization operator $P$. For the experiments carried out in this paper we decided to choose the $\ell_1$ norm[4]:

$$R(\chi) \approx \lambda_\chi \sum_{\substack{1 \leq k \leq d \\ 1 \leq i \leq n}} |\chi_i^k|,$$

with $\lambda_\chi \in \mathbb{R}$ being an hyper-parameter that measures the strength of the regularization.

In a deep architecture, when stacking multiple layers of kernel-based units, the non-monotonicity of the activation functions implies the absence of guarantees about the interval on which these functions operate, thus requiring them to be responsive to very heterogeneous inputs. In order to face this problem and to allow kernel-based units to concentrate their representational power on limited input ranges, it is possible to apply a normalization (Ioffe & Szegedy, 2015) to the input of the function. In particular, given $f_i(a_i^\kappa)$, $a_i^\kappa$ can be normalized as:

$$\hat{a}_i^\kappa = \gamma_i \frac{(a_i^\kappa - \mu_i)}{\sigma_i} + \beta_i; \qquad \text{where} \qquad \mu_i = \frac{1}{N} \sum_{\kappa=i}^N a_i^\kappa, \qquad \sigma_i = \frac{1}{N} \sum_{\kappa=i}^N (a_i^\kappa - \mu_i)^2;$$

while $\gamma_i$ and $\beta_i$ are additional trainable parameters.

## 4 EXPERIMENTS

We carried out several experiments in different learning settings to investigate the effectiveness of the KBDNN with emphasis on the adoption of kernel-based units in recurrent networks for capturing long-term dependences. Clearly, KBDNN architectures require to choose both the graph and the activation function. As it will be clear in the reminder of this section, the interplay of these choices leads to gain remarkable properties.

### 4.1 THE XOR PROBLEM.

We begin presenting a typical experimental set up in the classic XOR benchmark. In this experiment we chose a single unit with the Green function $g(z) = |z|$, so as $y = f(w_1 z_1 + w_2 z_2 + b)$ turns out to be

$$y = \sum_{k=1}^d \chi^k |w_1 z_1 + w_2 z_2 + b - c^k|$$

where $w_1, w_2$ and $b$ are trainable variables and the learning of $f$ corresponds with the discovery of both the centroids $c^k$ and the associated weights $\chi^k$. The simplicity of this learning task allows

---

[4]This choice is due to the fact that we want to enforce the sparseness of $\chi$, i.e. to use the smallest number of terms in expansion 6.

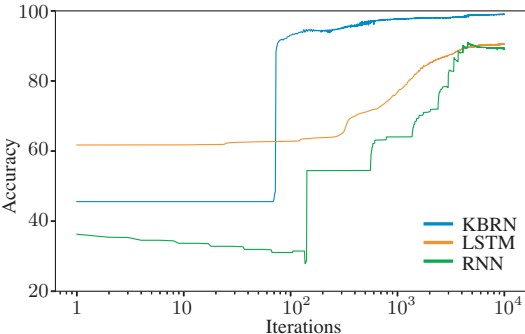

Figure 3: **Charging Problem.** The plot shows the accuracy obtained by recurrent nets with classic sigmoidal unit, LSTM cell, and KB unit. The horizontal axis is in logarithmic scale.

us to underline some interesting properties of KBDNNs. We carried out experiment by selecting a number of points for the expansion of the Green function that ranges from 50 to 300. This was done purposely to assess the regularization capabilities of the model, that is very much related to what typically happens with kernel machines. In Figure 2, we can see the neuron function $f$ at the end of the learning process under different settings. In the different columns, we plot function $f$ with a different numbers $d$ of clusters, while the two rows refer to experiments carried out with and without regularization. As one could expect, the learned activation functions become more and more complex as the number of clusters increases. However, when performing regularization, the effect of the kernel-based component of the architecture plays a crucial role by smoothing the functions significantly.

## 4.2 THE CHARGING PROBLEM

Let us consider a dynamical system which generates a Boolean sequence according to the model

$$
\begin{aligned}
h_t &= x_t + [h_{t-1} - 1 > 0] \cdot (h_{t-1} - 1) \\
y_t &= [h_t > 0],
\end{aligned}
\tag{7}
$$

where $h_{-1} = 0$, $x = \langle x_t \rangle$ is a sequence of integers and $y = \langle y_t \rangle$ is a Boolean sequence, that is $y_t \in \{0, 1\}$. An example of sequences generated by this system is the following:

$$
\begin{aligned}
t &= 0\ 1\ 2\ 3\ 4\ 5\ 6\ 7\ 8\ 9\ 10 \ldots \\
x_t &= 0\ 0\ 0\ 4\ 0\ 0\ 0\ 0\ 0\ 0\ 0 \ldots \\
y_t &= 0\ 0\ 0\ 1\ 1\ 1\ 1\ 0\ 0\ 0\ 0 \ldots.
\end{aligned}
$$

Notice that the system keeps memory when other 1 bit are coming, that is

$$
\begin{aligned}
t &= 0\ 1\ 2\ 3\ 4\ 5\ 6\ 7\ 8\ 9\ 10 \ldots \\
x_t &= 0\ 0\ 0\ 4\ 0\ 2\ 0\ 0\ 0\ 0\ 0 \ldots \\
y_t &= 0\ 0\ 0\ 1\ 1\ 1\ 1\ 1\ 1\ 0\ 0 \ldots
\end{aligned}
$$

The purpose of this experiment was that of checking what are the learning capabilities of KBRN to approximate sequences generated according to Eq. 7. The intuition is that a single KB-neuron is capable to *charge* the state according to an input, and then to *discharge* it until the state is reset. We generated sequences $\langle x_t \rangle$ of length $L = 30$. Three random element of each sequence were set with a random number ranging from 0 to 9. We compared KBRN, RNN with sigmoidal units, and recurrent with LSTM cells, with a single hidden unit. We used a KBRN unit with $d = 20$ centers to approximate the activation function. The algorithm used for optimization used the Adam algorithm with $\lambda = 0.001$ in all cases. Each model was trained for 10000 iterations with mini-batches of size 500. Figure 3 shows the accuracy on a randomly generated test set of size 25000 during the training process. The horizontal axis is in logarithmic scale.

## 4.3 LEARNING LONG-TERM DEPENDENCIES

We carried out a number of experiments aimed at investigating the capabilities of KBRN in learning tasks where we need to capture long-term dependencies. The difficulties of solving similar problems

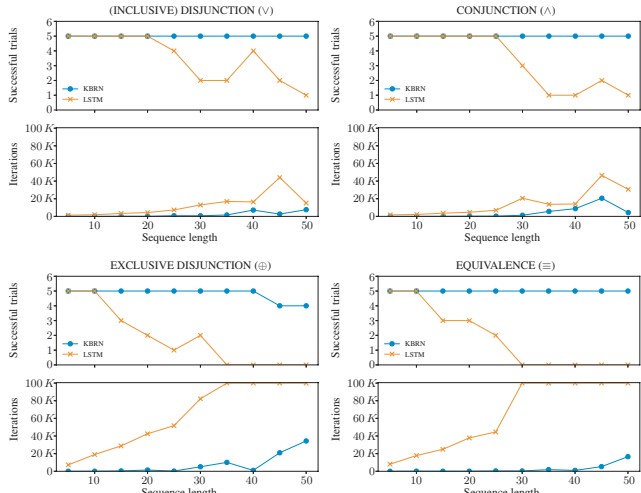

Figure 4: **Capturing Long-Term dependencies.** Number of successful trials and average number of iterations for a classification problem when the $\vee$, $\wedge$, $\oplus$ and $\equiv$ functions are used to determine the target, given the first two discriminant bits.

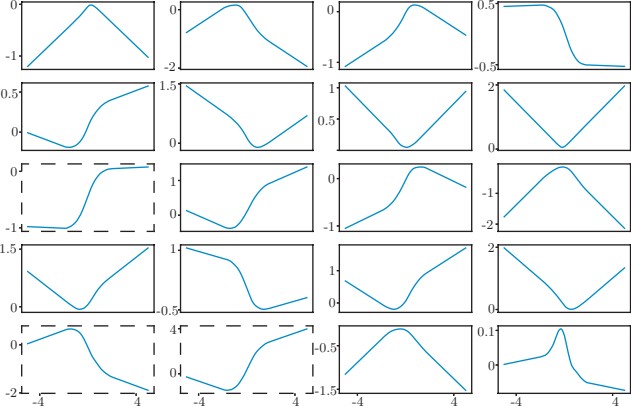

Figure 5: **Activation functions.** The 20 activation functions corresponding to the problem of capturing long-term dependencies in sequences that are only discriminated by the first two bit ($\equiv$ function). All functions are plotted in the interval $[-4, 4]$. The functions with a dashed frame are the ones for which $|f'| > 1$ in some subset of $[-4, 4]$.

was addressed in (Bengio et al., 1993) by discussions on gradient vanishing that is mostly due to the monotonicity of the activation functions. The authors also provided very effective yet simple benchmarks to claim that classic recurrent networks are unable to classify sequences where the distinguishing information is located only at the very beginning of the sequence; the rest of the sequence was supposed to be randomly generated. We defined a number of benchmarks inspired by the one given in (Bengio et al., 1993), where the decision on the classification of sequence $\langle x_t \rangle$ is contained in the first $L$ bits of a Boolean sequence of length $T \gg L$. We compared KBRN and recurrent nets with LSTM cells using an architecture where both networks were based on 20 hidden units. We used the Adam algorithm with $\lambda = 0.001$ in all cases. Each model was trained for a maximum of $100{,}000$ iterations with mini-batches of size $500$; for each iteration, a single weight update was performed. For the LSTM cells, we used the standard implementation provided by TensorFlow (following (Zaremba et al., 2014)). For KBRN we used a number of centroids $d = 100$ and the described normalization.

We generated automatically a set of benchmarks with $L = 2$ and variable length $T$, where the binary sequences $\langle x_t \rangle$ can be distinguished when looking simply at the first two bits, while the the rest is

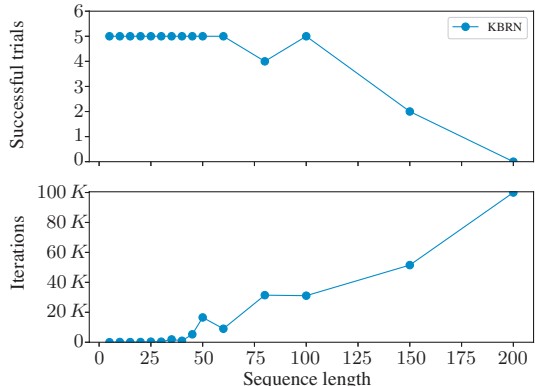

Figure 6: **Capturing Long-Term dependencies.** Number of successful trials and average number of iterations when facing the $\equiv$ problem with sequences of length ranging from 5 to 200, when the distinguishing information is located in the first two bits.

a noisy string with uniformly random distribution. Here we report some of our experiments when choosing the first two discriminant bits according to the $\vee$, $\wedge$, $\oplus$ and $\equiv$ functions.

For each Boolean function, that was supposed to be learned, and for several sequence lengths (up to 50), we performed 5 different runs, with different initialization seeds. A trial was considered successful if the model was capable of learning the function before the maximum allowed number of iterations was reached. In Figure 4 we present the results of these experiments. Each of the four quadrants of Figure 4 is relative to a different Boolean function, and reports two different plots. The first one has the sequence length on the $x$-axis and the number of successful trials on the $y$-axis. The second plot has the sequence length on the $x$-axis and, on the $y$-axis, the average number of iterations required to solve the task. The analysis of these plots allows us to draw a couple of interesting conclusions: *(i)* KBRN architectures are capable of solving the problems in almost all cases, regardless of the sequence length, while recurrent networks with LSTM cells started to experiment difficulties for sequences longer than 30, and *(ii)*, whenever convergence is achieve, KBRN architectures converge significantly faster than LSTM.

In order to investigate with more details the capabilities of KBRN of handling very long sequences, we carried out another experiment, that was based on the benchmark that KBRN solved with more difficulty, namely the equivalence ($\equiv$) problem. We carried out a processing over sequences with length $60, 80, 100, 150$, and $200$. In Figure 6, we report the results of this experiment. As we can see, KBRN are capable of solving the task even with sequences of length 150, eventually failing with sequences of length 200.

## 5 CONCLUSIONS

In this paper we have introduced Kernel-Based Deep Neural Networks. The proposed KBDNN model is characterized by the classic primal representation of deep nets, that is enriched with the expressiveness of activation functions given by kernel expansion. The idea of learning the activation function is not entirely new. However, in this paper we have shown that the KBDNN representation turns out to be the solution of a general optimization problem, in which both the weights, that belong to a finite-dimensional space, and the activation function, that are chosen from a functional space are jointly determined. This bridges naturally the powerful representation capabilities of deep nets with the elegant and effective setting of kernel machines for the learning of the neuron functions.

A massive experimentation of KBDNN is still required to assess the actual impact of the appropriate activation function in real-world problems. However, this paper already proposes a first important conclusion which involves recurrent networks, that are based on this kind of activation function. In particular, we have provided both theoretical and experimental evidence to claim that the KBRN architecture exhibits an ideal computational structure to deal with classic problems of capturing long-term dependencies.

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
