# OpenReview forum: "Learning Neuron Non-Linearities with Kernel-Based Deep Neural Networks"
_ICLR.cc/2019/Conference_

### Official Review · AnonReviewer2 · 2018-10-30
**Learning Neuron Non-Linearities with Kernel-Based Deep Neural Networks**

**Rating:** 6
**Confidence:** 4

**Review:**

The scope of the paper is interesting: to additionally learn the nonlinear activation function of the neuron.

The insights provided in section 2 with eqs (2)-(5) are interesting and naturally build on the previous work of Poggio & Girosi (1990) and Smola (1998). I found this a nice new insight and the strongest part of the paper. It is e.g. revealing to see to which P and L the rectifier nonlinearity is corresponding.

On the other hand I also have a number of suggestions for further improvement:

- Section 1: related to the overall function to be learned, the authors state "this general problem has been already solved". I think this statement is not completely correct, because depending on the choice of the stabilizer one obtains different optimal representations (e.g. Gaussian RBF or thin plate splines) as explained in Poggio & Girosi (1990). The theory does not tell what the best stabilizer is.

Additional relevant work that would be good to mention at this point, in the area of kernel methods, is e.g. learning the kernel.

- It seems that no other existing work on deep kernel machines has been mentioned in the paper, while in the conclusions the authors state "In this paper we have introduced Kernel-based deep neural networks".

- Related to the training set T_N the notation e^kappa is not explained. It is not clear how this is related to eq (1).

- It would be good to comment on the difference between (3)(4) and Poggio & Girosi (1990).

- unnumbered eq after (5): are there multiple solutions to the problem (non-convex)?

- The explanation of the recurrent network at the end of section 2 is too limited. Moreover, LSTM is not just a neuron nonlinearity, but a recurrent network with a particular structure. To which P and L would LSTM correspond?

- Fig.2: some of the nonlinearities look quite complicated and some of them are oscillatory (is this desirable? it reminds us of overfitting). Often one is interested in activation functions with a "simple shape" like sigmoid, tanh, relu. A more complicated nonlinearity may reduce the interpretability of the model.

- The examples given are rather conceptual (though nice) examples of the proposed method. However, no comparisons with other methods have been made yet in terms of generalization performance, e.g. on a few standard classification benchmark data sets, in comparison with other deep or shallow models.

A possible drawback of the proposed method might be (or maybe not) that additional unknown parameters need to be learned, which could possibly lead to worse generalization. It might be good to further investigate this.

---

### Official Review · AnonReviewer3 · 2018-11-03
**Unclear presentation and weak results.**

**Rating:** 4
**Confidence:** 3

**Review:**

Summary: the authors propose a method for learning activation functions in neural networks using kernels. Each activation function is modeled as a weighted set of kernels, where (as I understand it) the weights are learned simultaneously with the linear weights in the network. The authors apply their method to learn the XOR function and to a simple sequence memory task.

My main concern with the paper is that the presentation is very hard to follow. The motivation, background, and contributions of this paper are all mixed in the abstract and introduction, making it hard to understand what is the current state of learned activations, what this paper introduces, and how the work in this paper relates to prior work. Instead, I found the presentation in Scardapane et al much clearer (that paper has a clear separation of background, related work, and their contributions). By contrast, this paper has one large paragraph in the introduction that muddles together multiple threads of thought, making it hard to digest. Before the reader has had time to digest the main ideas, the paper launches into a highly technical description of the method, without a clear high level explanation of what the main technique is. A lot of the mathematical notation introduced in Sec2 is not clearly motivated.

My other main concern is with the results. The paper solves two simple tasks with their method, and it is unclear what their kernel methods really buy them. For the XOR problem, it is such a simple task that it is hard to judge the method (it seems like complex activation functions are not required to solve the task, and it is not clear what including them gets you). For the sequence memory task, it seems unfair to compare their results to recurrent networks with a *single* hidden unit. If you have networks with 2 or 4 or 8 hidden units, do they solve the task? These experiments do not shed much light on the advantages of using kernel activations in recurrent networks.

---

### Official Review · AnonReviewer1 · 2018-11-05
**LEARNING NEURON NON-LINEARITIES WITH KERNEL-BASED DEEP NEURAL NETWORKS**

**Rating:** 5
**Confidence:** 3

**Review:**

The paper investigates the problem of designing the activation functions of neural networks with focus on recurrent architectures. The authors frame such problem as learning the activation functions in the space of square integrable functions by adding a regularization term penalizing the differential properties of candidate functions. In particular, the authors observe that this strategy is related to some well-established approaches to select activation functions such as ReLUs.


The paper has some typos and some passages are hard to read/interpret. The write-up needs to be improved significantly.

While some of the observations reported by the authors are interesting it is in general hard to evaluate the contributions of the paper. In particular the discussion of Sec. 2 is very informal, although ti describes the key technical observations used in the paper to devise the model (Sec. 3) that is then evaluated in the experiments (Sec. 4). In particular, it is unclear whether the authors are describing some known results - in which case they should add references - or original contributions - in which case they should report their results with more mathematical rigour. Indeed, in the abstract, the authors state that a representation theorem is given, but in the text they provide only an informal discussion of such result.

Overall, it is hard to agree with the authors' conclusion that "the KBRN architecture exhibits an ideal computational structure to deal with classic problems of capturing long-term dependencies": the theoretical discussion does not provide sufficient evidence in this sense.

Some minor points:

Confusing notation: why were the alpha^k replaced with the \chi^k between Sec. 2 and Sec. 3?

Unclear motivation for some design choices. For instance 1) the justification given by the authors to neglect the linear terms from both g(x) and k(x) in Sec. 3 is unclear. 2) why was the \ell_1 norm used as penalty for the regularizer R(\chi) in Sec.3? One could argue that \ell_1 is used to encourage sparse solutions, but the authors should explain why sparsity is desirable in this setting.

---

### Meta-Review · Area_Chair1 · 2018-12-13
**An interesting proposal to automatically learn activation non-linearities but novelty is unclear and the paper is hard to follow.**

**Confidence:** 5
**Recommendation:** Reject

**Metareview:**

This paper proposes to automatically learn the form of the non-linearities of neural networks in deep neural networks, which the reviewers noted to be an interesting albeit significantly studied direction.   Overall, this paper falls just below the bar, with no reviewer really willing to champion for acceptance.  Reviewer 3 found the paper to be marginally above the acceptance threshold and found the insights provided in the paper (in Section 2) to be a neat and strong contribution.  Reviewers 1-2, however, found the paper marginally below the bar and seemed confused by the presentation of the paper.  They seemed to believe in the motivation and idea, but they found the paper hard to follow and not particularly clearly written.  It would seem that the paper could significantly benefit from careful editing and restructuring to disambiguate contributions from motivation and existing literature.  Also, the authors should provide clear justification for their design choices and modeling assumptions.